Ecological and Evolutionary Science

# Complementary Roles of Wood-Inhabiting Fungi and Bacteria Facilitate Deadwood Decomposition

Vojtěch Tláskal,[a,b] Vendula Brabcová,[a] Tomáš Větrovský,[a] Mayuko Jomura,[c] Rubén López-Mondéjar,[a]
Lummy Maria Oliveira Monteiro,[d] João Pedro Saraiva,[d] Zander Rainier Human,[a] Tomáš Cajthaml,[a] Ulisses Nunes da Rocha,[d]
🔟 Petr Baldrian[a]

[a]Institute of Microbiology of the Czech Academy of Sciences, Prague, Czech Republic
[b]Faculty of Science, Charles University, Prague, Czech Republic
[c]Department of Forest Science and Resources, College of Bioresource Sciences, Nihon University, Fujisawa, Kanagawa, Japan
[d]Department of Environmental Microbiology, UFZ-Helmholtz Centre for Environmental Research, Leipzig, Germany

**ABSTRACT** Forests accumulate and store large amounts of carbon (C), and a substantial fraction of this stock is contained in deadwood. This transient pool is subject to decomposition by deadwood-associated organisms, and in this process it contributes to $CO_2$ emissions. Although fungi and bacteria are known to colonize deadwood, little is known about the microbial processes that mediate carbon and nitrogen (N) cycling in deadwood. In this study, using a combination of metagenomics, metatranscriptomics, and nutrient flux measurements, we demonstrate that the decomposition of deadwood reflects the complementary roles played by fungi and bacteria. Fungi were found to dominate the decomposition of deadwood and particularly its recalcitrant fractions, while several bacterial taxa participate in N accumulation in deadwood through N fixation, being dependent on fungal activity with respect to deadwood colonization and C supply. Conversely, bacterial N fixation helps to decrease the constraints of deadwood decomposition for fungi. Both the $CO_2$ efflux and N accumulation that are a result of a joint action of deadwood bacteria and fungi may be significant for nutrient cycling at ecosystem levels. Especially in boreal forests with low N stocks, deadwood retention may help to improve the nutritional status and fertility of soils.

**IMPORTANCE** Wood represents a globally important stock of C, and its mineralization importantly contributes to the global C cycle. Microorganisms play a key role in deadwood decomposition, since they possess enzymatic tools for the degradation of recalcitrant plant polymers. The present paradigm is that fungi accomplish degradation while commensalist bacteria exploit the products of fungal extracellular enzymatic cleavage, but this assumption was never backed by the analysis of microbial roles in deadwood. This study clearly identifies the roles of fungi and bacteria in the microbiome and demonstrates the importance of bacteria and their N fixation for the nutrient balance in deadwood as well as fluxes at the ecosystem level. Deadwood decomposition is shown as a process where fungi and bacteria play defined, complementary roles.

**KEYWORDS** bacteria, deadwood, decomposition, forest ecosystems, fungi, metatranscriptomics, microbiome, nitrogen fixation, nutrient cycling

Address correspondence to Petr Baldrian, baldrian@biomed.cas.cz.

🐦 Fungi and bacteria play complementary roles in deadwood transformation where fungi dominate in decomposition while bacteria and their N2 fixation are important for the nutrient balance in deadwood as well as fluxes at the ecosystem level.

Forests play a crucial role in making the Earth habitable by maintaining biodiversity and serving as an important part of biogeochemical cycles (1, 2). Forests, especially unmanaged natural forests, accumulate and store large amounts of carbon (3). A substantial fraction of this C stock, 73 ± 6 Pg, or 8% of the total global forest C stock (2), is contained within deadwood. This C pool is transient, because during its transformation by saprotrophic organisms, most C is liberated as $CO_2$ into the atmosphere, while the

rest is sequestered in soils as dissolved organic C or within microbial biomass along with other nutrients. Recent reports estimate the annual deadwood C efflux through respiration from temperate and boreal stands to be 0.5 to 3.6 Mg C ha$^{-1}$ year$^{-1}$ (4, 5). Thus, the C efflux originating in deadwood decomposition is comparable to that of forest soil respiration of 7.8 and 5.0 Mg C ha$^{-1}$ year$^{-1}$ in temperate and boreal forests, respectively (6).

Deadwood is a specific substrate that is rich in C but highly recalcitrant and physically impermeable. Deadwood hosts a wide range of fungi and bacteria (7, 8), and the cord-forming basidiomycetes are considered major wood decomposers due to their strong enzymatic production. This feature makes them able to degrade all important components of wood and to rapidly penetrate it (9, 10). Furthermore, fresh deadwood of most temperate and boreal trees has a low nitrogen content, ranging between 0.03 and 0.19% of dry mass (11), which represents a major limitation for decomposition (12). During decomposition, the N content in deadwood typically increases (13, 14) as a consequence of C loss by respiration. In addition, bacterial fixation of atmospheric $N_2$ was shown to substantially contribute to the N increase in deadwood during decomposition (15, 16). N fixation is highly energy-demanding, and young deadwood with available C sources represents a potentially ideal setting for this process (17). In specific situations, N content in deadwood also may be increased through translocation by ectomycorrhizal or certain other fungi (9, 16). We ask how C utilization and N cycling in deadwood are coupled and what the roles are of the individual members of deadwood microbiome. The answers to these questions are especially important for the understanding of the ecology of high-latitude forests that are typically N limited and both symbiotic and asymbiotic soil N fixation is low (18, 19). Under these conditions, N fixation in deadwood may represent a process of ecosystem-level importance.

In this study, we utilized the combination of metagenomics, metatranscriptomics, and gas flux measurements to analyze the participation of the deadwood-associated microbiome of a European beech (*Fagus sylvatica*)-dominated temperate natural forest in the cycling of C and N. Our findings suggest that deadwood in the ecosystem of temperate forests is a hot spot of bacterial N fixation and fungal degradation. We demonstrate that decomposition of deadwood is done through complementing traits of fungi and bacteria playing specific roles in C and N cycling.

## RESULTS AND DISCUSSION

None of the parameters of wood chemistry were significantly different between old and young deadwood (Fig. 1). In particular, the wood density, indicating the progress of decomposition, showed high variation, ranging from 290 to 470 kg m$^{-3}$. This is not surprising, since individual deadwood logs are under the complex influence of the individual history of the microbial community assembly and microclimate conditions, including sun exposure and soil contact (20, 21). Both $CO_2$ production from deadwood and N fixation showed a significant correlation with pH ($r = -0.720$, $P = 0.016$ and $r = -0.832$, $P = 0.003$, respectively). The young and old deadwood had similar levels of respiration of $1.43 \pm 0.31$ and $1.21 \pm 0.38$ g $CO_2$ kg$^{-1}$ day$^{-1}$ (Fig. 1), comparable to estimates from other temperate forests (4, 5). Considering that the average deadwood stock in the studied natural forest is 208 m³ ha$^{-1}$ (22), the $CO_2$ flux from deadwood would be in a similar range of $CO_2$ flux from soils in temperate and boreal forests, $7.8 \pm 10.4$ and $5.0 \pm 18.0$ Mg ha$^{-1}$ year$^{-1}$, respectively (6). This confirms previous reports of deadwood decomposition as an important $CO_2$ source (5).

The rates of N fixation were significantly higher in young deadwood than in old deadwood ($297 \pm 103$ and $37 \pm 20$ ng N g$^{-1}$ day$^{-1}$, $P = 0.031$) (Fig. 1). These results reflect that the N content is very low in fresh *Fagus sylvatica* deadwood, while it is high in labile C sources that are required to fuel this highly energy-consuming process (17). As decomposition proceeds, N accumulation relieves N limitation in deadwood and, thus, affects microbial composition at the end of the deadwood life cycle. The accumulated N represents an input into forest soils with a potentially high impact, especially in

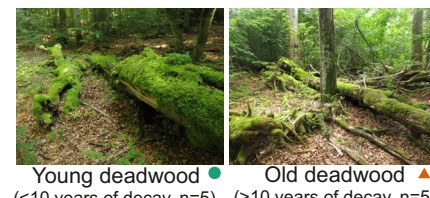

| | Young deadwood ● | Old deadwood ▲ |
|---|---|---|
| | (<10 years of decay, n=5) | (>10 years of decay, n=5) |
| density (kg m$^{-3}$) | 397 ± 30 | 376 ± 24 |
| C content (%) | 44.9 ± 5.6 | 49.0 ± 0.3 |
| N content (%) | 0.53 ± 0.11 | 0.36 ± 0.04 |
| lignin content (%) | 28.8 ± 4.7 | 26.5 ± 3.2 |
| pH | 4.0 ± 0.2 | 4.4 ± 0.2 |
| bacterial 16S (10$^9$ copies g$^{-1}$) | 1.38 ± 0.32 | 1.20 ± 0.32 |
| fungal 18S (10$^9$ copies g$^{-1}$) | 2.05 ± 0.40 | 1.35 ± 0.44 |
| fungal 18S / bacterial 16S | 2.00 ± 0.51 | 1.66 ± 0.82 |
| CO$_2$ production (g kg$^{-1}$ day$^{-1}$) | 1.43 ± 0.31 | 1.21 ± 0.38 |
| N$_2$ fixation (μg kg$^{-1}$ day$^{-1}$) | 297 ± 103 | 37 ± 20 * |

**FIG 1** Comparison of properties of young and old deadwood. Chemical properties and microbial biomass across samples of old and young deadwood and the rates of $CO_2$ production and N fixation. Due to high variation, none of the measured parameters was significantly different among young and old deadwood, except for the N fixation rate, which was significantly higher in young deadwood (*). The values indicate means and standard errors ($n = 5$); N fixation was estimated in an acetylene reduction assay.

N-limited high-latitude forests (18). The area-approximated N fixation of 0.49 g $N_2$ m$^{-2}$ year$^{-1}$ for the studied forest at the optimal temperature of 25°C is comparable to reports from boreal forests (15, 16). These values are typically higher than those reported for asymbiotic N fixation in soils of temperate and boreal forests (23, 24), which underlines the importance of N fixation in deadwood as the source of new N input (25).

Decomposing deadwood appeared to be rich in microbial biomass irrespective of deadwood age, with the number of copies of bacterial and fungal ribosomal DNA being approximately 10$^9$ copies per g dry mass, and the counts of fungi were slightly higher than those of bacteria (Fig. 1). The composition of total biomass based on the small subunit rRNA, representing the pool of ribosomes, identified fungi and bacteria as the dominant components of the deadwood microbiome, with Arthropoda and Nematoda representing the most abundant nonmicrobial organisms. This finding also confirmed the dominance of Eukaryota, which represented 50 to 94% of the rRNA pool (Fig. 2a). Identified transcripts were again mostly assigned to Eukaryota (58 to 94%), with most of them being affiliated with the Basidiomycota and Ascomycota fungi; on average, the share of fungal transcripts was 85% and that of bacteria was 13% (Fig. 2b). When looking at the expression of ribosomal proteins that may represent a proxy of growth (26), the share of bacteria was slightly higher (Fig. 2c).

Importantly, the fungal community and transcriptional activity in all but three decomposing logs were dominated by Basidiomycota, where they were responsible for 72.5% ± 1.0% of transcription (Fig. 2b). In three logs, Ascomycota showed the highest share of transcription (38.6% ± 2.1%), and in this instance, the share of bacterial transcription was also higher (27.7% ± 3.5%), being approximately 4-fold higher than that of logs dominated by Basidiomycota (Fig. 2b). This distinction in microbiome composition is likely due to the colonization history of the substrate (20) but may also reflect the antagonism of Basidiomycota toward bacteria (7).

Deadwood is theoretically an unlimited stock of organic carbon, and its utilization is the main source of energy and biomass for deadwood-associated organisms. The share of genes with a known role in the decomposition of plant, fungal, and bacterial biomass ranged between 0.11% and 0.61% of the total transcription. This value did not differ among old and young deadwood but was significantly lower in the Ascomycota-dominated logs than in the Basidiomycota-dominated logs (0.12 ± 0.01 and 0.42 ± 0.06%,

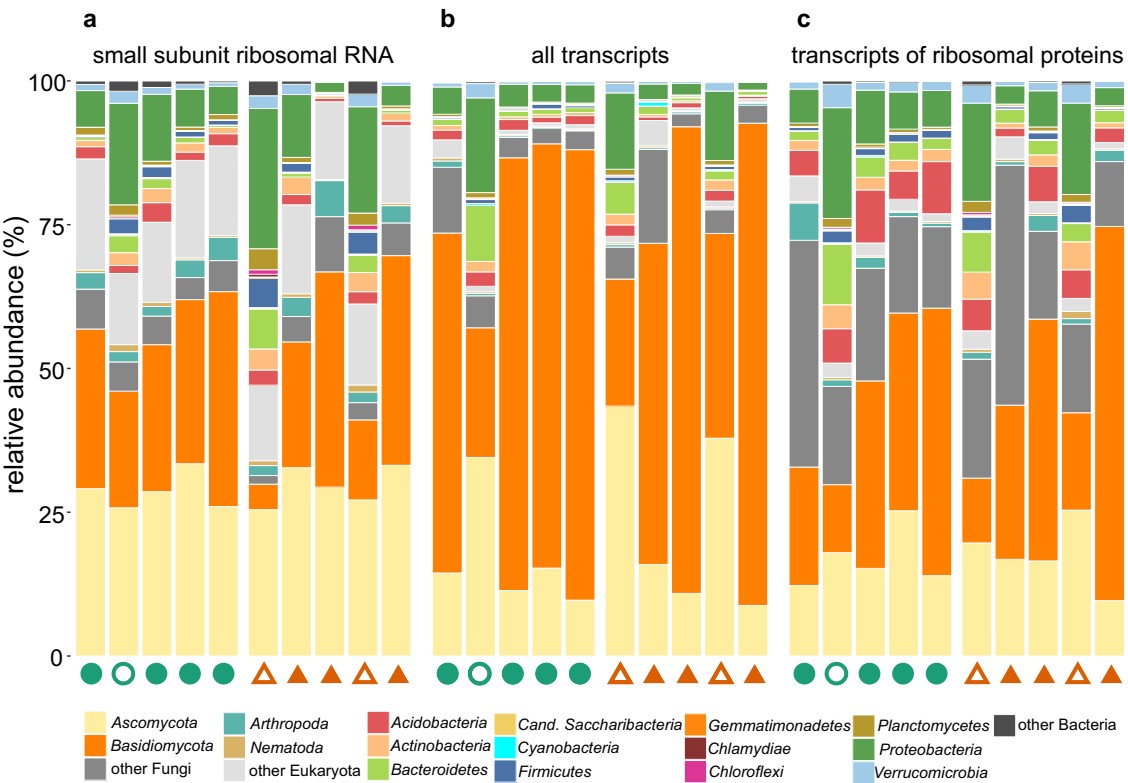

**FIG 2** Taxonomic composition and activity of deadwood-associated organisms. Composition of the community of deadwood-associated organisms based on the relative abundance of small ribosomal subunit RNA, corresponding to ribosome counts (a), their activity based on the relative abundance of all mRNA transcripts (b), and their growth based on the relative abundance of mRNA of genes encoding ribosomal proteins (c). Filled symbols indicate samples dominated by Basidiomycota, and open symbols indicate those rich in Ascomycota and bacteria.

respectively, $P = 0.018$) (Fig. 3a). Enzyme expression also significantly decreased with pH ($r = -0.927$, $P < 0.001$). The pool of carbohydrate-active enzymes (CAZymes) was mostly composed of those targeting cellulose (0.17% of all transcripts), cellobiose and xylobiose (0.042%), fungal glucans (0.031%), lignin (0.020%), and chitin (0.015%) (Fig. 3a). The CAZyme families participating in the degradation of cellulose (AA9, GH5, and GH7), lignin (AA2), and chitin (GH18) were among the most abundant in both the metagenome and metatranscriptome (see Fig. S1 in the supplemental material), indicating that plant and fungal biomass are the most important C sources (Fig. 3a). Ordination of samples based on the CAZyme composition showed separation of Basidiomycota- and Ascomycota-dominated logs as well as clustering by deadwood age (Fig. 3b).

As much as 91% of transcripts of CAZymes decomposing biopolymers in deadwood were assigned to fungi with a share of bacteria of only 7% (Fig. 3c). Although fungi are also major producers of CAZymes in other environments, contributing, for instance, 78% and 42% in forest litter and soil (27, 28), the extent of fungal dominance in CAZyme transcription in deadwood is unprecedented (Fig. S1). For the two most recalcitrant wood biopolymers, lignin and cellulose, the ratio of fungal to bacterial transcripts was as high as 642:1. While 29.8% of CAZymes produced by fungi targeted lignin or cellulose, only 1.4% of CAZymes produced by bacteria targeted these substrates. Since the degradation of these structural biopolymers is required for wood colonization, fungi appear to play a decisive role in opening the physically recalcitrant substrate for utilization by deadwood-associated biota. In this context, the role of Basidiomycota should be stressed, since they represent the only microbial phylum to degrade lignin (Fig. S1). The bacterial contribution to C utilization was higher in the cases of starch/glycogen and peptidoglycan (Fig. 3c and Fig. S1). Methylotrophy was

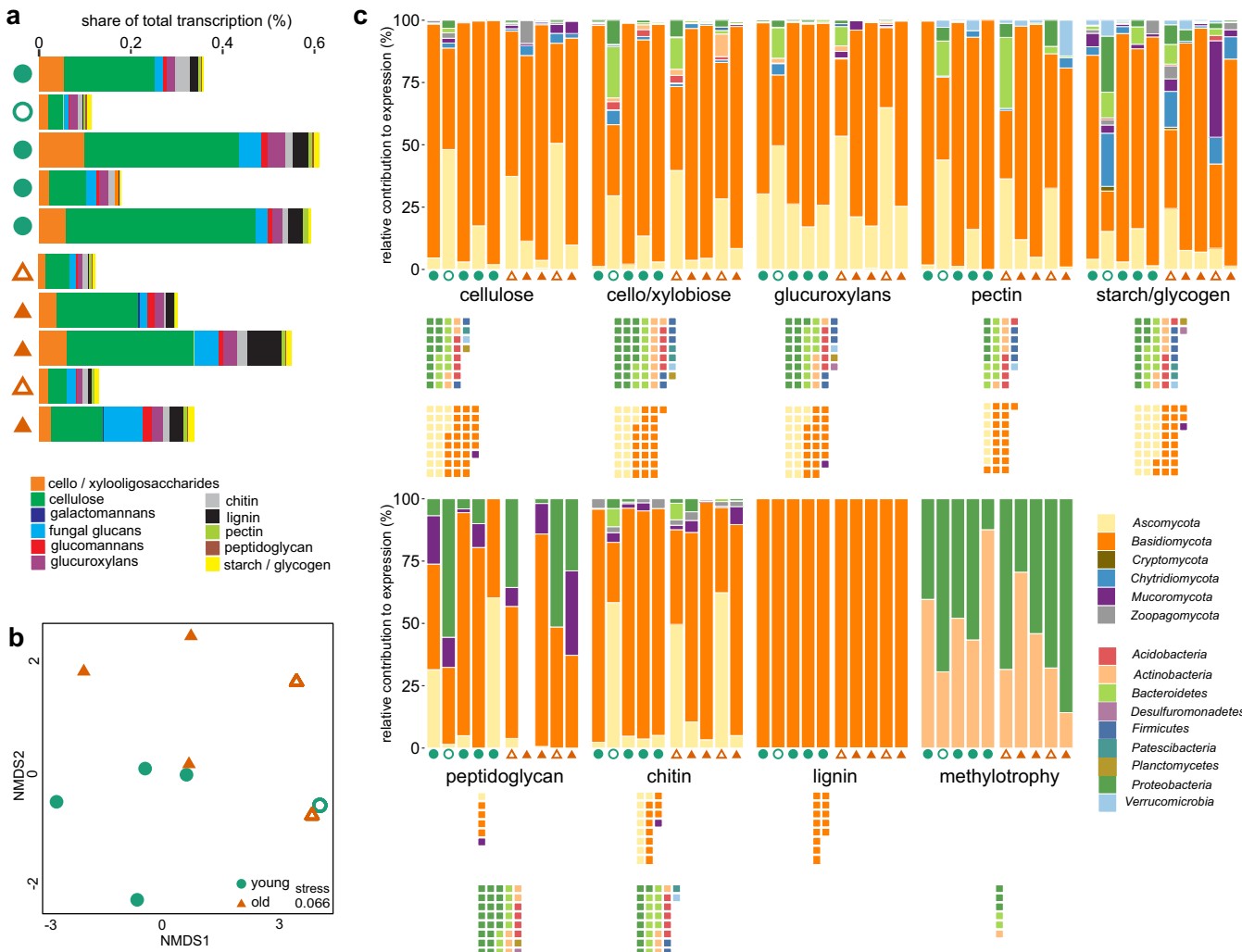

**FIG 3** Functional diversity of enzymes targeting sources of carbon. The functional and taxonomic breakdown of CAZymes and methylotrophy genes is shown. The relative rate of transcription of CAZymes (GH and AA families) targeting relevant wood substrates across samples (a) and two-dimensional NMDS based on Euclidean distance of the transcription rates of individual CAZy families (b). (c) Relative contribution of fungi and bacteria to the expression of CAZymes and methylotrophy genes by target. Squares indicate the diversity and taxonomic assignment of fungal genera and bacterial MAGs with the potential to utilize a given substrate. Each square corresponds to one bacterial MAG or one fungal genus.

present in *Proteobacteria* and *Actinobacteria*, although its contribution to C utilization was relatively small, with 24 ± 3 reads per million reads being observed (Fig. 3c). In this bacterium-specific pathway, methylotrophs utilize C from methane and methanol, the by-products of fungal wood decomposition (29).

N-cycling genes accounted for an average of 0.06% of transcripts. Of these transcripts, genes involved in ammonia incorporation into organic molecules represented 85.8% and were transcribed by all microbial taxa (Fig. 4), catalyzing the recycling of N liberated from organic compounds. In agreement with the high rates of potential N fixation, N fixation was the dominant process of the transformation of inorganic N with a share of 7.4% of all N-cycling genes (Fig. 4). Excluding ammonia assimilation, the nitrogenase *nifD* was the most transcribed gene overall in the N cycle. N fixation was mostly performed by *Proteobacteria* but also by *Firmicutes*, *Chlorobi*, *Verrucomicrobia*, and *Bacteroidetes*. Nitrate and nitrite reduction were represented by 3.5% and 2.8% of N-cycling transcripts, with dissimilatory steps in the N cycle being represented by only 1.0% (Fig. 4). Compared to soils (30), the respiratory pathways utilizing nitrate or nitrite seem to be considerably less important in deadwood. Importantly, missing nitrification steps and high rates of ammonia assimilation underscore that almost all available N is

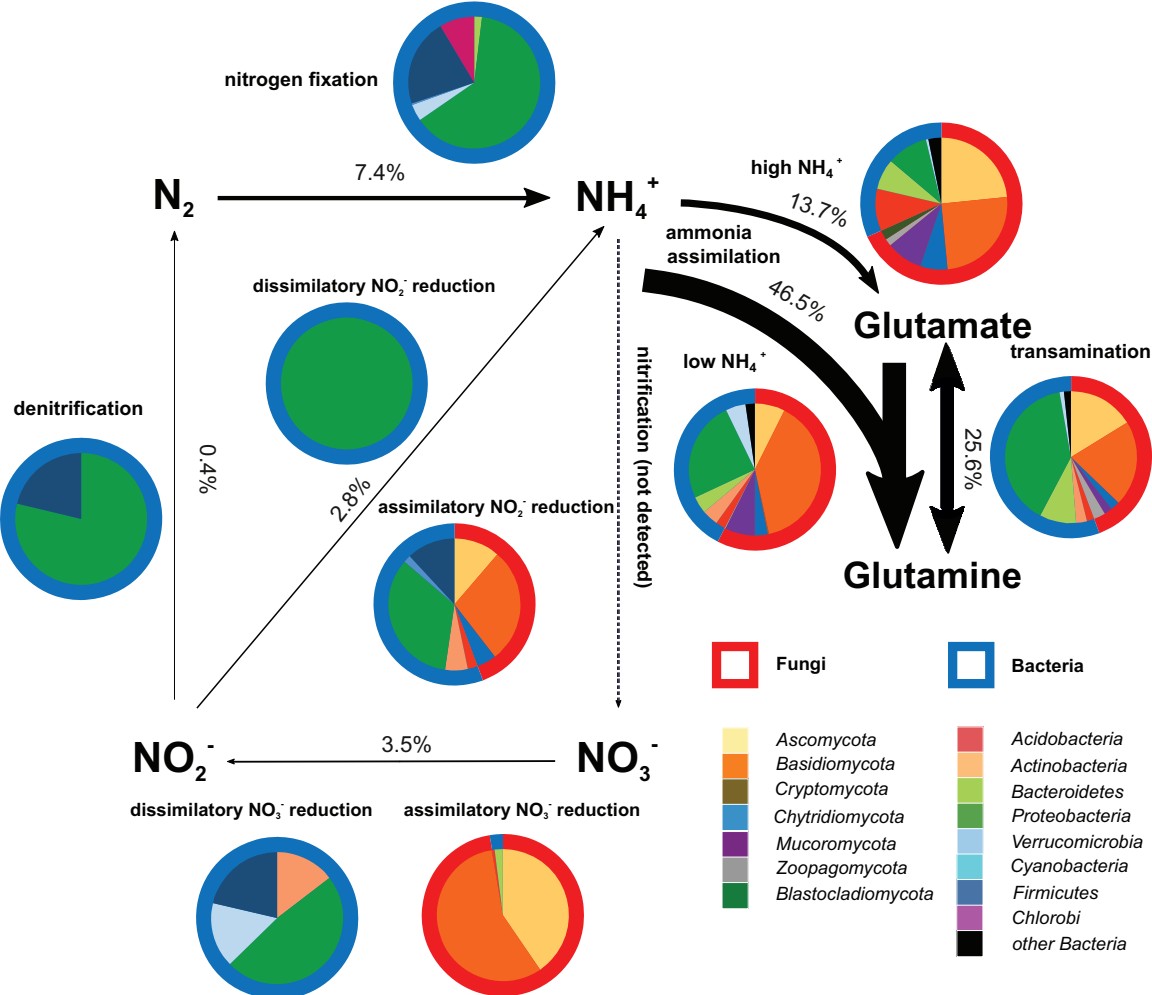

**FIG 4** Nitrogen cycle and ammonia assimilation pathways occurring in deadwood. N-cycling intensity is expressed as the relative share of expression of each function across all N-cycling genes. Pie charts indicate the relative shares of bacterial and fungal transcription in each process. Nitrification was not detected.

incorporated into living systems. This finding is, again, in contrast with soils where large parts of N are oxidized by nitrification of bacteria to gain energy (27, 31).

The assignment of genes and transcripts to 58 high-quality metagenome-assembled genomes (MAGs) of bacteria or to fungal taxa makes it possible to find adaptations of microorganisms to life on decomposing deadwood (Fig. S2 and S4). The ability of bacteria to utilize a more or less wide range of C compounds showed phylogenetic clustering, with the MAGs classified as *Actinobacteria* and *Proteobacteria* containing 36 ± 5 and 46 ± 5 CAZymes per genome compared to 101 ± 15 and 122 ± 16 in the most versatile *Acidobacteria* and *Bacteroidetes* (Fig. S3). Several bacterial genomes (members of *Acidobacteria*, *Bacteroidetes*, *Firmicutes*, *Proteobacteria*, and *Planctomycetes*) combined the potential to fix N and to produce CAZymes targeting dominant plant biopolymers, such as cellulose, which makes them independent in C and N utilization (Fig. S2). Other N-fixing taxa within *Actinobacteria* and *Proteobacteria* lack the ability to decompose complex biopolymers and have to rely on the utilization of cello- and xylooligosaccharides provided by other decomposers. As the last option, the members of the order *Rhizobiales* combine N fixation with methylotrophy (Fig. S2), being dependent on the supply of C by fungal wood decomposers (32).

The classification of fungal transcripts indicates that within Basidiomycota, white-

rot fungi capable of lignin degradation, such as *Armillaria*, *Ganoderma*, or *Phlebia*, play important roles in deadwood decomposition. The dominant members of Ascomycota, such as *Xylaria* and *Anthostoma*, belong to soft-rot fungi that preferentially utilize cellulose and hemicelluloses in deadwood. The spectra of utilized C biopolymers across fungi are relatively wide, and cellulose is decomposed by most of the dominant taxa (Fig. S4). Fungal involvement in the N cycle is often constrained to the incorporation of ammonia into biomass, with assimilatory pathways on nitrate and nitrite reduction playing a much more limited role (Fig. S4). The fungal component of the deadwood microbiome is relatively diverse and includes, among others, plant pathogens and ectomycorrhizal fungi associated with tree seedlings that germinate on deadwood (Fig. S4). Among nutritional guilds, white-rot fungi display the most diverse enzyme sets, being able to utilize various carbon sources, while members of other guilds are less versatile. Only one of three dominant ectomycorrhizal fungi expressed polysaccharide-targeting enzyme-degrading fungal glucans (Fig. S5), which is not surprising, considering the low occurrence of biopolymer-degrading genes in their genomes (33). The versatility of white-rot basidiomycetes, enabling them to rapidly colonize wood by degrading both lignin and cellulose, is the likely reason for their high abundance and functional dominance in the majority of decomposing logs. Moreover, rapid hyphal growth was found to be a key predictor of overall deadwood decomposition rate, showing that versatile groups have direct influence on this process (34).

While deadwood potentially represents an exclusive stock of C and fungi often behave antagonistically toward competitors to capture as much of the deadwood resources as possible (7, 35), their activity in deadwood also opens a window of opportunity for selected bacteria (36). The colonization of deadwood by fungal hyphae promotes the spread of bacteria that can slide along hyphae (37) or benefit from nutrient redistribution (38). Moreover, the high share of expression of cello- and xylooligosaccharide-degrading enzymes by bacteria, such as $\beta$-glucosidases, indicates their utilization of cellulose and xylan fragments, the products of (almost exclusively) fungal degradation of the recalcitrant biopolymers. Previous analyses of deadwood microbiomes suggested cooperation between N-fixers and fungi (39, 40). While some bacterial N-fixers may be independent in C acquisition, methylotrophic and commensalist N-fixers depend on products of biopolymer decomposition by fungi. On the fungal side, N accumulation in deadwood due to bacterial fixation promotes the succession of decomposers. The white-rot fungi parasitic on living trees and primary colonizers of deadwood are adapted to N-limited conditions, such as *Fomitopsis*, with only 0.8% N in mycelia (41). These fungi, however, are not able to completely utilize deadwood resources. Many later colonizers have larger N requirements, with N content in mycelia of up to 6.4% as in the case of *Lycoperdon* (41). Although some N can be scavenged from dead mycelia of primary colonizers (42), bacterial N fixation alleviates N limitation and allows the complete recycling of organic matter by late decomposers and ultimately the incorporation of residual material into soil. Thus, the role partitioning between bacteria and fungi seems to be important for complete deadwood decomposition. In specific situations, such as in *Picea abies* forests where young trees establish on decomposing deadwood, N may be translocated to deadwood from soil by ectomycorrhizal or soil-foraging fungi (16). However, the *Fagus sylvatica* deadwood studied here was devoid of tree seedlings and ectomycorrhizal fungi (Fig. S4), and no overlap of cord-forming fungi was found between deadwood and the soil beneath it (43).

While the partitioning of roles between fungi and bacteria appears to be important for the individual fates of decomposing deadwood logs, the whole process also has ecosystem-level consequences. $CO_2$ fluxes from deadwood appear to be quantitatively important in forests where deadwood stocks are large. Although more research is still needed, it seems that the amount of fixed $N_2$ during deadwood decomposition may represent an important source of N entering the soil environment at the end of the deadwood life cycle. Especially in boreal forests with low N stocks, deadwood retention may help to improve the nutritional status and fertility of forest soils.

## MATERIALS AND METHODS

**Study site.** The study was performed in the Žofínský Prales National Nature Reserve, an unmanaged forest in the south of the Czech Republic (48°39'57"N, 14°42'24"E). The core zone of the forest reserve (42 ha) had never been managed and any human intervention stopped in 1838, when it was declared as a reserve. Thus, it represents a rare fragment of European temperate virgin forest with deadwood left to spontaneous decomposition. The reserve is situated at 730 to 830 m above sea level, and the bedrock is almost homogeneous and consists of fine to medium-grainy porphyritic and biotite granite. Annual average rainfall is 866 mm, and annual average temperature is 6.2°C (44). At present, the reserve is covered by a mixed forest where *Fagus sylvatica* predominates in all diameter classes (51.5% of total living wood volume), followed by *Picea abies* (42.8%) and *Abies alba* (4.8%). The mean living tree volume is 690 m$^3$ h$^{-1}$, and the volume of coarse woody debris (logs, represented by tree trunks and their fragments, from 102 to 310 m$^3$ h$^{-1}$ with an average of 208 m$^3$ h$^{-1}$) (22, 45). Logs are repeatedly surveyed, and an approximate age of each log, the cause of death (e.g., stem breakage, windthrow, etc.), and status before downing (fresh or decomposed) is known (13).

**Study design and sampling.** Previous analysis indicated that deadwood age (time of decomposition) significantly affects both wood chemistry and the composition of microbial communities (13, 46). Thus, we randomly selected dead tree trunks that represented young deadwood (<10 years of decomposition, $n = 5$) and old deadwood (>10 years of decomposition, $n = 5$); only trees that were not alive and not decomposed before downing were considered. Selection was enabled by a database of deadwood with data from historical surveys. For metagenome (MG) analysis, to obtain a more comprehensive annotation source, 15 additional logs were sampled, giving a total 25 metagenomic samples. Sampling was performed in November 2016. The length of each selected log (or the sum of the lengths of its fragments) was measured, and four samples were collected evenly along the log length by drilling vertically from the middle of the upper surface through the whole diameter using an electric drill with an auger diameter of 10 mm. The sawdust from all four samples from each log was pooled and immediately frozen using liquid nitrogen, transported to the laboratory on dry ice, and stored at −80°C until further processing. For the measurement of $CO_2$ production and N fixation, compact pieces of deadwood of an approximate volume of 4 cm$^3$ (three replicates per log) were collected and brought to the laboratory under refrigeration to be used for immediate analysis.

**Deadwood chemistry, $CO_2$ production, and N fixation.** For pH measurement, approximately 0.5 g of subsampled wood material was mixed with 5 ml of deionized water, left overnight at 4°C, and shaken for 60 min on an orbital shaker before assessment. The C and N contents were measured in an external laboratory: a freeze-dried subsample was milled (<0.1 mm particle size), and 10 to 30 mg of material was burned in 100% $O_2$ at 1,000°C. The amounts of oxidized C and N were recorded by Flash 2000 (Thermo Scientific) and analyzed with Eager Xperience software (Thermo Scientific) to obtain their relative contents. Deadwood density was estimated as a ratio of its dry mass after freeze-drying and volume.

Three independent samples of compact wood material per log, each of an approximate volume of 10 cm$^3$, were collected for the estimation of $CO_2$ production and N fixation rates according to reference 5. For $CO_2$ production estimation, samples were incubated for 10 min in an acrylic chamber (volume, 2.2 liters). $CO_2$ concentration and humidity in the chamber were monitored with an infrared gas analyzer GMP 343 (Vaisala Inc.) and hydrometer RTR 503 (T and D Inc.). Per-second data were recorded by a data logger GL200A (Graphtec Inc.). The temperature of the surface of the samples was measured using a Fluke 561 infrared thermometer (Fluke Inc.). The last 5 min of incubation were used to calculate the respiration rate to avoid the effect of moisture change. To adjust for mean temperature at the site (6.2°C), the model $R_{CWD} = a \times e^{bT}$ was used to transform respiration data (4). For N fixation, an acetylene reduction assay was performed by following reference 47. Ten percent of the headspace of the 4-ml vial with wood block was replaced with acetylene gas and incubated at the temperature of maximal nitrogenase activity, 25.2°C, for 24 h (18). The concentration of ethylene in the headspace was analyzed by a gas chromatograph (436-Scion; Bruker). Head space (100 $\mu$l) of the samples was used for injection, and the analytes were separated isothermally at 30°C using a Restek Rt U-Bond 30-m column (inner diameter, 0.25 mm; film thickness, 8 $\mu$m), equipped with a PLOT column particle trap (2.5 m). The injector temperature was set to 240°C with a 1:20 splitter, and the carrier gas was helium (5.0; 2 ml min$^{-1}$). A conversion factor from acetylene reduction to N fixation of 4 was used (15). For retrieval of global records of soil $CO_2$ efflux, the SRDB database was accessed in February 2020 (git version 8ab37bf, temperate forest records $n = 1,310$, boreal forests records $n = 198$ [6]).

**Nucleic acid extraction.** Wood samples (approximately 10 g of material) were homogenized using a mortar and pestle under liquid nitrogen prior to nucleic acid extraction and thoroughly mixed. Total DNA was extracted in triplicate from 200-mg batches of finely ground wood powder using a NucleoSpin soil kit (Macherey-Nagel).

Total RNA was extracted in triplicate from 200-mg batches of sample using a NucleoSpin RNA plant kit (Macherey-Nagel) according to the manufacturer's protocol after mixing with 900 $\mu$l of the RA1 buffer and shaking on FastPrep-24 (MP Biomedicals) at 6.5 ms$^{-1}$ twice for 20 s. Triplicates were pooled and treated with a OneStep PCR inhibitor removal kit (Zymo Research). DNA was removed using a DNA-free DNA removal kit (Thermo Fisher Scientific). The efficiency of DNA removal was confirmed by the negative PCR results with the bacterial primers 515F and 806R (48). RNA quality was assessed using a 2100 Bioanalyzer (Agilent Technologies).

**Analysis of deadwood-associated organisms.** To estimate the relative representation of bacteria, fungi, and other organisms in deadwood, reads of small subunit rRNA, representing the ribosome counts, were identified and classified from total RNA. Libraries for high-throughput sequencing of total

RNA were prepared using a TruSeq RNA sample prep kit v2 (Illumina) according to the manufacturer's instructions, omitting the initial capture of poly(A) tails to enable total RNA to be ligated. Samples were pooled in equimolar volumes and sequenced on an Illumina HiSeq 2500 with a rapid run $2\times 250$ option at Brigham Young University Sequencing Centre, USA. Raw FASTQ files were processed using Trimmomatic 0.36 (49) and FASTX-Toolkit (http://hannonlab.cshl.edu/fastx_toolkit/) to filter out adaptor contamination, trim low-quality ends of reads, and omit reads with overall low quality ($<$30). Sequences shorter than 40 bp were omitted. Non-rRNA reads were filtered out from the files using the bbduk.sh 38.26 program in BBTools (https://sourceforge.net/projects/bbmap/). BLASTn 2.7.1+ was used against the manually curated database SilvaMod, derived from Silva nr SSU Ref v128 (50, 51), to infer the best 50 hits with output as full-alignment XML file for all reads in each sample. Furthermore, the LCAClassifier script from CREST software (50) was used to construct alignments to find the best possible classification of each read with the lowest common ancestor approach.

Bacterial and fungal rRNA gene copies in DNA samples were quantified by quantitative PCR (qPCR) using the 1108f and 1132r primers for bacteria (52, 53) and FR1 and FF390 primers for fungi (54) as described previously (55). The fungal/bacterial rRNA gene ratio was calculated by dividing rRNA gene copy numbers.

**Metagenomics and metatranscriptomics.** For metatranscriptome (MT) analysis, the content of rRNA in RNA samples was reduced as described previously (55) using a combination of Ribo-Zero rRNA removal kit human/mouse/rat and Ribo-Zero rRNA removal kit bacteria (Illumina). Oligonucleotide probes annealed to rRNA from both types of kits were mixed together and added to each sample. The efficiency of the removal was checked using a 2100 Bioanalyzer, and removal was repeated when necessary. Reverse transcription was performed with SuperScript III (Thermo Fisher Scientific). Libraries for high-throughput sequencing were prepared using the ScriptSeq v2 RNA-Seq library preparation kit (Illumina) according to the manufacturer's instructions with a final 14 cycles of amplification by FailSafe PCR enzyme (Lucigen).

The NEBNext Ultra II DNA library prep kit for Illumina (New England BioLabs) was used to generate metagenome libraries according to the manufacturer's instructions. Samples of the metagenome and metatranscriptome were pooled in equimolar volumes and sequenced on an Illumina HiSeq 2500 with a rapid-run $2\times 250$ option at Brigham Young University Sequencing Centre, USA.

MG assembly and annotation were performed as described previously (27). Briefly, Trimmomatic 0.36 (49) and FASTX-Toolkit (http://hannonlab.cshl.edu/fastx_toolkit/) were used to remove adaptor contamination, trim low-quality ends of reads, and omit reads with overall low quality ($<$30); sequences shorter than 50 bp were omitted. The combined assembly of all 25 samples was performed using MEGAHIT 1.1.3 (56). Metagenome sequencing yielded on average 22.5 $\pm$ 7.2 million reads per sample that were assembled into 17,936,557 contigs over 200 bp in length.

MT assembly and annotation were performed as described previously (55). Trimmomatic 0.36 (49) and FASTX-Toolkit (http://hannonlab.cshl.edu/fastx_toolkit/) were used to remove adaptor contamination, trim low-quality ends of reads, and omit reads with overall low quality ($<$30); sequences shorter than 50 bp were omitted. mRNA reads were filtered from the files using the bbduk.sh 38.26 program in BBTools (https://sourceforge.net/projects/bbmap/). Combined assembly was performed using MEGAHIT 1.1.3 (56). Metatranscriptome sequencing yielded, on average, 31.3 $\pm$ 9.1 million reads per sample that were assembled into 1,332,519 contigs over 200 bp in length.

Gene calling was performed using MG-RAST (57) and yielded a total of 22,171,460 and 1,404,953 predicted coding regions for MG and MT, respectively. Taxonomic identification was performed in MG-RAST as well as using BLAST against all published fungal genomes available in January 2018 (58). Of these two, taxonomic identification with a higher bit score was used as the best hit. Functions of predicted genes were annotated with the *hmmsearch* function in HMMER 3.2.1 (59) using the FOAM database as a source of HMMs for relevant genes (60). Genes encoding the carbohydrate-active enzymes (CAZymes) were annotated using the dbCAN HMM database V6 (61). CAZymes were grouped based on their participation in the utilization of distinct C sources based on their classification into known CAZyme families (62). Genes active in methylotrophy and N cycling were assigned to individual processes based on their KEGG classification (Tables S1, S2, and S3).

Taxonomic and functional annotations with E values higher than $10E^{-30}$ were disregarded. The contribution of various organisms to total transcription was based on the relative abundances of all reads (Fig. 2b) and reads of genes encoding ribosomal proteins (Fig. 2c). Since ribosomes are produced during cell division, these values should correspond to the growth of the organisms (26). Metatranscriptomic reads with the best hit to fungi were classified to fungal genera. Fungal genera were classified into the following ecophysiological groups: white rot, brown rot, soft rot, other saprotrophs, plant pathogens, animal parasites, mycoparasites, and ectomycorrhizal fungi based on published literature, with the definitions of the groups being the same as those in reference 63.

**Identification and analysis of metagenome-assembled genomes.** The creation and analysis of metagenome-assembled genomes make it possible to analyze the traits of individual prokaryotic taxa without the need of their isolation and, thus, are more representative for the study of ecosystem processes (64). Bins that represent prokaryotic taxa present in the metagenome were constructed using MetaBAT2 (65) with default settings, except for the minimal length of contigs set to 2,000 bp, which produced bins with overall better statistics than the minimal 2,500-bp size. CheckM 1.0.11 (66) served for assigning taxonomy and statistics to bins with the *lineage_wf* pipeline. Bins with a completeness score greater than 50% were selected for quality improvement using RefineM according to the instructions of the developers (67). Briefly, scaffolds with genomic properties (GC content, coverage profiles, and tetranucleotide signatures) whose values were different from those expected in each bin were excluded.

These values were calculated based on the mean absolute error and correlation criteria. Next, the refined bins were further processed to identify and remove scaffolds with taxonomic assignments different from those assigned to the bin. Lastly, the scaffolds that possessed 16S rRNA genes divergent from the taxonomic affiliation of the refined bins were removed. The taxonomy of the bins was inferred by GTDB-Tk (68). Fifty-eight bins with quality scores of >50 (CheckM completeness value minus 5× contamination value) were considered metagenome-assembled genomes (MAGs) as defined by reference 67 and retained for analysis.

Metagenomic and metatranscriptomic reads were mapped to MAGs as described previously (26) using bowtie 2.2.4 (69) to infer the abundance of bins across samples and the expression of bin-specific genes. Relative abundances of mapped reads were obtained by dividing the number of mapped reads by the number of all reads in a given sample.

**Statistical analyses.** Statistical analysis was performed in the R environment (70). Two-dimensional nonmetric multidimensional scaling (NMDS) was used to plot the variation in the relative abundance of reads of the glycoside hydrolase and auxiliary enzyme families of the CAZymes using the function *metaMDS* from the vegan package (71) and Euclidean distance of the Hellinger-transformed CAZy abundance. Analysis of variance (ANOVA) was used to test for the differences in environmental variables. Spearman correlation coefficients were used as a measure of relationships between variables, and Monte Carlo permutation tests were used to determine the *P* values for correlations. Figures were generated using custom Java and R scripts (70), Rstudio (72), Tidyverse (73), and Inkscape (https://inkscape .org/).

**Data availability.** The code for reproducing sequence processing is provided at https://github.com/ TlaskalV/Deadwood-microbiome. Data described in the manuscript, including raw sequences (total RNA, metatranscriptome, and metagenome), assembly files, and resolved MAGs, have been deposited in NCBI under BioProject accession number PRJNA603240. Properties of dead *Fagus sylvatica* logs analyzed in this paper are provided in Table S4 in the supplemental material.

## SUPPLEMENTAL MATERIAL

Supplemental material is available online only.

**FIG S1**, PDF file, 1.3 MB.

**FIG S2**, PDF file, 1.3 MB.

**FIG S3**, PDF file, 0.3 MB.

**FIG S4**, PDF file, 1.3 MB.

**FIG S5**, PDF file, 1.3 MB.

**TABLE S1**, DOCX file, 0.02 MB.

**TABLE S2**, DOCX file, 0.01 MB.

**TABLE S3**, DOCX file, 0.02 MB.

**TABLE S4**, DOCX file, 0.01 MB.

## ACKNOWLEDGMENTS

This work was supported by the Czech Science Foundation (17-20110S), by the Ministry of Education, Youth and Sports of the Czech Republic (LTT17022), and by Charles University (GAUK 950217). Computational resources were supplied by the project "e-Infrastruktura CZ" (e-INFRA LM2018140), provided within the program Projects of Large Research, Development and Innovations Infrastructures.

We have no competing interests to declare.

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
