## [Reviewer comments · mSystems]

Complementary roles of wood-inhabiting fungi and bacteria facilitate deadwood decomposition

Vojtěch Tláškal, Vendula Brabcová, Tomáš Větrovský, Mayuko Jomura, Rubén López-Mondéjar, Lummy Maria Oliveira Monteiro, Joao Pedro Saraiva, Zander Human, Tomáš Cajthaml, Ulisses Nunes da Rocha, and Petr Baldrian

Corresponding Author(s): Petr Baldrian, Institute of Microbiology of the Czech Academy of Sciences

Review Timeline:

Submission Date:	October 16, 2020
Editorial Decision:	November 13, 2020
Revision Received:	November 23, 2020
Accepted:	December 13, 2020

Editor: Karoline Faust

Reviewer(s): Disclosure of reviewer identity is with reference to reviewer comments included in decision letter(s). The following individuals involved in review of your submission have agreed to reveal their identity: Yahya Kooch (Reviewer #1); Maraike Probst (Reviewer #2)

Transaction Report:

DOI: <https://doi.org/10.1128/mSystems.01078-20>

November 13, 2020

Dr. Petr Baldrian
Institute of Microbiology of the Czech Academy of Sciences
Laboratory of Environmental Microbiology
Víteňská 1083
Praha 4 14220
Czech Republic

Re: mSystems01078-20 (Deadwood decomposition is facilitated by the complementarity of roles of wood-inhabiting fungi and bacteria)

Dear Dr. Petr Baldrian:

The reviewers have now returned their assessments. While I am pleased to inform you that both of them agree that minor modifications will be sufficient, I ask you to carefully address the remaining concerns. Please also make sure to share all sequencing data and sample metadata (BioProject PRJNA603240 is not yet accessible).

Below you will find the comments of the reviewers.

To submit your modified manuscript, log onto the eJP submission site at <https://msystems.msubmit.net/cgi-bin/main.plex>. If you cannot remember your password, click the "Can't remember your password?" link and follow the instructions on the screen. Go to Author Tasks and click the appropriate manuscript title to begin the resubmission process. The information that you entered when you first submitted the paper will be displayed. Please update the information as necessary. Provide (1) point-by-point responses to the issues raised by the reviewers as file type "Response to Reviewers," not in your cover letter, and (2) a PDF file that indicates the changes from the original submission (by highlighting or underlining the changes) as file type "Marked Up Manuscript - For Review Only."

Due to the SARS-CoV-2 pandemic, our typical 60 day deadline for revisions will not be applied. I hope that you will be able to submit a revised manuscript soon, but want to reassure you that the journal will be flexible in terms of timing, particularly if experimental revisions are needed. When you are ready to resubmit, please know that our staff and Editors are working remotely and handling submissions without delay. If you do not wish to modify the manuscript and prefer to submit it to another journal, please notify me of your decision immediately so that the manuscript may be formally withdrawn from consideration by mSystems.

Sincerely,

Karoline Faust

Editor, mSystems

Journals Department
Reviewer comments:

Reviewer #1 (Comments for the Author):

This manuscript deals with deadwood decomposition is facilitated by the complementarity of roles of wood-inhabiting fungi and bacteria. The topic fits well with the journal aims and scope and it would be of interest for its readers, however, some of weaknesses are detected in this manuscript as below:

- (1) The innovation of this study is absent.
- (2) The new references are absent in whole of text.
- (3) The discussion is mainly descriptive, with little explanation on the new references behind the observed results.

Reviewer #2 (Comments for the Author):

General comments

The ecological role of bacteria and fungi in deadwood decomposition is a field of ongoing research, as their contribution to the complex process is still not well understood - and depending on numerous environmental factors. The study "Deadwood decomposition is facilitated by the complementarity of roles of wood-inhabiting fungi and bacteria" addresses this issue in a straightforward, question-oriented way. The authors sampled *Fagus sylvatica* deadwood logs of two different age classes (old (> 10 years) and young deadwood (< 10 years)) from a natural reserve, which is a forest that is untouched for many years. They measured the deadwood logs' respective abiotic properties, analysed nutrient fluxes and sequenced both their metagenomes and metatranscriptomes. A general problem in deadwood research is the slow speed of the process, which disables to observe the same log over the whole process of decay. Although this is always a drawback, it does not weaken the study at hand. The authors confirmed fungi as the most abundant and important deadwood degraders, which were associated to the C cycle. Bacteria contributed to the process by fixing vital N, which is limiting the degradation process. The authors show that both, CO₂ and N₂ flux were a product of joint, interactive and complementary fungal and

bacterial action. In addition, the authors conclude that in the deadwood ecosystem, which eventually turns to soil, C and N for the most part are stored in biomass, although there are substantial differences between both ecosystems. Consequently, they suggest that especially in boreal forests, deadwood retention may help to improve the nutritional status and fertility of soils.

I much liked the manuscript. The story is well written, the aim is clear and sound and the experimental design follows the rationale. The results are presented in a focused, straightforward manner, which is easy to follow. The figures underline the text well. The joint discussion makes it easy to follow and answer the pending questions. In my opinion, this manuscript clarifies some ongoing questions, provides useful, interesting information and is a sound foundation for future research on the topic. I liked the distinction between basidiomycete and ascomycete dominated logs and the implication of this fungal colonization. The authors present a very clear pattern, which from my perspective could be emphasized and elaborated on a bit more.

-

Specific comments

The terms decay, degradation and decomposition are used synonymously. In one section (line 37), transformation is used. I recommend sticking to one term to avoid confusion. In the literature, all terms are used. In my opinion, an exact terminology can help sharpening the content.

Nitrogen and N are not used consistently. Please stick to one option. The same is true for N-fixing and N fixing (and N-fixers vs. N fixers). Please homogenize throughout the manuscript.

Please insert a space between value and {degree sign}C.

Please abbreviate day by d.

Title

I dislike the term complementarity. Although its meaning is clear and I do like neologisms, I discourage their use in the title. What about an active form: "Complementary roles of wood-inhabiting fungi and bacteria facilitate deadwood decomposition"?

Abstract

Line 23: "associate with" does not sound right to me. I would use colonize.

Introduction

Lines 61-64: this sentence is long and appears a bit wordy. I recommend splitting it. What about: "Deadwood hosts a wide range of fungi and bacteria (7,8). Cord-forming basidiomycetes are considered the major decomposers. The enzymes they produce enable rapid penetration of the substrate and decomposition of all complex wood components (9,10)."

Lines 72-73: The structure of the sentence is awkward (are in the end). Please rephrase.

Line 75: "while"? If I understand correctly, the conjunction should be "and". If I understand correctly, I would phrase "...are typically N-limited and have low rates of symbiotic and asymbiotic soil N-fixation."

Lines 77-78: I would mention the aims and design before the results. Consequently, I would move this sentence at the very end of the introduction.

Results and discussion

Line 101: Please remove "still". It implies a process (over time), which I found confusing as a state is referred to.

Line 103: "succession at the end of the deadwood lifecycle". I find this confusing. If I understand correctly, I would phrase "composition". Then I would start a new sentence.

"As decomposition proceeds, N accumulation relieves N limitation and thus affects deadwood microbial composition. At the end of the deadwood lifecycle, the accumulated N represents an input

into forest soils with a potentially high impact, especially in N-limited high latitude forests (18)."

Line 110: I would use "appeared" instead of "was" (rich in microbial biomass).

Lines 151-152: Fungal phyla cannot be discriminated in Fig. S1. I recommend including this finding a bit better into the manuscript.

Line 160: "all deadwood microbes". This is not necessarily true. I would say by "transcribed by all microbial taxa".

Line 167: There is a comma missing after "In comparison to soils (31)". Please insert.

Line 221: The abbreviation ECM is introduced in the methods section, which comes afterwards. Please include its introduction in the results (if you want to abbreviate at all).

Material and methods

In this section, hectar appears abbreviated with h instead of ha, which was confusing at first. Please clarify.

Metagenomes:

- If I understand correctly, a number of $5 + 15 = 20$ logs were sampled. DNA was extracted in triplicates. Were sixty metagenomes sequenced? In line 341, a number of 25 samples are mentioned. I am confused by this number. Please clarify and include the total number of metagenome samples.

- For metagenomes, the library preparation is described in the section "Nucleic acid extraction". For metatranscriptomes, the library preparation is mentioned in the section "Metagenomics and metatranscriptomics". Please homogenize.

The qPCR is not included in the methods. Please insert.

Statistical data analysis: I am puzzled by using square-root transformed data and Euclidean distance for the composition of enzyme families of CAZymes. Especially in combination with a PERMANOVA on bray Curtis distance, this seems a bit inconsistent to me. Can you explain, please?

In this regard, would you mind providing values on the MDS-axes, please (Fig. 2B)?

Figures and tables

Figure 2, caption: I suggest removing "other microbes" and give "Ascomycota and bacteria" directly; no need for information in brackets.

Figure S1: Maybe this is me, but it took me a moment to understand that the bar and square colors are the same both rows... Could you move up both legends and indicate "bars" and "squares" as legend title, please?

Authors response to reviewers comments

Authors responses are in bold script

The reviewers have now returned their assessments. While I am pleased to inform you that both of them agree that minor modifications will be sufficient, I ask you to carefully address the remaining concerns. Please also make sure to share all sequencing data and sample metadata (BioProject PRJNA603240 is not yet accessible).

PRJNA603240 released on 2020/11/19

Reviewer comments:

Reviewer #1 (Comments for the Author):

This manuscript deals with deadwood decomposition is facilitated by the complementarity of roles of wood-inhabiting fungi and bacteria. The topic fits well with the journal aims and scope and it would be of interest for its readers, however, some of weaknesses are detected in this manuscript as below:

(1) The innovation of this study is absent.

We believe that the paper contains multiple innovative aspects, although it partly confirms hypotheses expressed in the past. For example, we clearly show the importance of N₂ fixation in N cycling and indicate bacterial taxa (MAGs) that perform it, while previously these processes were only observed without link to bacterial taxa or just inferred based on the presence on bacteria of certain taxonomy, without a clear proof of their properties. Moreover, by demonstration of the dependence of several N-fixers on fungi, we provide the link between eukaryotic and prokaryotic processes. In agreement with Reviewer 2, we also believe that the distinction of logs dominated by the Basidiomycota and the Ascomycota is both novel and functionally important. This innovation is possible because the paper provides the first view of deadwood processes by means of metatranscriptome analysis.

(2) The new references are absent in whole of text.

New references were added where appropriate

(3) The discussion is mainly descriptive, with little explanation on the new references behind the observed results.

Results and discussion are merged in this paper and the results are thus discussed directly in the content. We have added new references where appropriate.

Reviewer #2 (Comments for the Author):

General comments

The ecological role of bacteria and fungi in deadwood decomposition is a field of ongoing research, as

their contribution to the complex process is still not well understood - and depending on numerous environmental factors. The study "Deadwood decomposition is facilitated by the complementarity of roles of wood-inhabiting fungi and bacteria" addresses this issue in a straightforward, question-oriented way. The authors sampled *Fagus sylvatica* deadwood logs of two different age classes (old (> 10 years) and young deadwood (< 10 years)) from a natural reserve, which is a forest that is untouched for many years. They measured the deadwood logs' respective abiotic properties, analysed nutrient fluxes and sequenced both their metagenomes and metatranscriptomes. A general problem in deadwood research is the slow speed of the process, which disables to observe the same log over the whole process of decay. Although this is always a drawback, it does not weaken the study at hand. The authors confirmed fungi as the most abundant and important deadwood degraders, which were associated to the C cycle. Bacteria contributed to the process by fixing vital N, which is limiting the degradation process. The authors show that both, CO₂ and N₂ flux were a product of joint, interactive and complementary fungal and bacterial action. In addition, the authors conclude that in the deadwood ecosystem, which eventually turns to soil, C and N for the most part are stored in biomass, although there are substantial differences between both ecosystems. Consequently, they suggest that especially in boreal forests, deadwood retention may help to improve the nutritional status and fertility of soils.

I much liked the manuscript. The story is well written, the aim is clear and sound and the experimental design follows the rationale. The results are presented in a focused, straightforward manner, which is easy to follow. The figures underline the text well. The joint discussion makes it easy to follow and answer the pending questions. In my opinion, this manuscript clarifies some ongoing questions, provides useful, interesting information and is a sound foundation for future research on the topic. I liked the distinction between basidiomycete and ascomycete dominated logs and the implication of this fungal colonization. The authors present a very clear pattern, which from my perspective could be emphasized and elaborated on a bit more.

-

Specific comments

The terms decay, degradation and decomposition are used synonymously. In one section (line 37), transformation is used. I recommend sticking to one term to avoid confusion. In the literature, all terms are used. In my opinion, an exact terminology can help sharpening the content.

Terms were unified: degradation is used in the substance context (lignin, cellulose) and decomposition is now used in the general context of the wood habitat.

Nitrogen and N are not used consistently. Please stick to one option. The same is true for N-fixing and N fixing (and N-fixers vs. N fixers). Please homogenize throughout the manuscript.

Nitrogen is now abbreviated throughout the manuscript as N and N-cycling genes, nitrogen fixation as N fixation. When mentioning atmospheric nitrogen, chemical formula N₂ is used.

Please insert a space between value and {degree sign}C.

Changed as requested

Please abbreviate day by d.

Changed as requested

Title

I dislike the term complementity. Although its meaning is clear and I do like neologisms, I discourage their use in the title. What about an active form: "Complementary roles of wood-inhabiting fungi and bacteria facilitate deadwood decomposition"?

We have accepted the suggested change of the title

Abstract

Line 23: "associate with" does not sound right to me. I would use colonize.

Changed as recommended

Introduction

Lines 61-64: this sentence is long and appears a bit wordy. I recommend splitting it. What about: "Deadwood hosts a wide range of fungi and bacteria (7,8). Cord-forming basidiomycetes are considered the major decomposers. The enzymes they produce enable rapid penetration of the substrate and decomposition of all complex wood components (9,10)."

This long sequence was now divided

Lines 72-73: The structure of the sentence is awkward (are in the end). Please rephrase.

Rephrased

Line 75: "while"? If I understand correctly, the conjunction should be "and". If I understand correctly, I would phrase "...are typically N-limited and have low rates of symbiotic and asymbiotic soil N-fixation."

Changed as suggested

Lines 77-78: I would mention the aims and design before the results. Consequently, I would move this sentence at the very end of the introduction.

The end of the introduction was modified based on this suggestion

Results and discussion

Line 101: Please remove "still". It implies a process (over time), which I found confusing as a state is referred to.

Changed based on this suggestion

Line 103: "succession at the end of the deadwood lifecycle". I find this confusing. If I understand correctly, I would phrase "composition". Then I would start a new sentence.

"As decomposition proceeds, N accumulation relieves N limitation and thus affects deadwood microbial composition. At the end of the deadwood lifecycle, the accumulated N represents an input into forest soils with a potentially high impact, especially in N-limited high latitude forests (18)."

Rephrased

Line 110: I would use "appeared" instead of "was" (rich in microbial biomass).

Changed as suggested

Lines 151-152: Fungal phyla cannot be discriminated in Fig. S1. I recommend including this finding a bit better into the manuscript.

We believe that distinguishing fungal phyla is not essential in this figure. While it would be doable for major fungal phyla, labelling of bacterial phyla would be too messy. We also believe that the important information on the contribution of fungal phyla to biopolymer decomposition is sufficiently covered in Figure 3.

Line 160: "all deadwood microbes". This is not necessarily true. I would say by "transcribed by all microbial taxa".

Corrected

Line 167: There is a comma missing after "In comparison to soils (31)". Please insert.

Comma inserted

Line 221: The abbreviation ECM is introduced in the methods section, which comes afterwards. Please include its introduction in the results (if you want to abbreviate at all).

ECM abbreviation replaced by the word ectomycorrhizal throughout the manuscript

Material and methods

In this section, hectar appears abbreviated with h instead of ha, which was confusing at first. Please clarify.

Abbreviation for hectare changed to ha

Metagenomes:

- If I understand correctly, a number of $5 + 15 = 20$ logs were sampled. DNA was extracted in triplicates. Were sixty metagenomes sequenced? In line 341, a number of 25 samples are mentioned. I am confused by this number. Please clarify and include the total number of metagenome samples.

Total number of MT and MG samples is now explicitly mentioned in the manuscript

- For metagenomes, the library preparation is described in the section "Nucleic acid extraction". For metatranscriptomes, the library preparation is mentioned in the section "Metagenomics and metatranscriptomics". Please homogenize.

Library preparation of the MG samples moved to the proper section Metagenomics and metatranscriptomics.

The qPCR is not included in the methods. Please insert.

qPCR methods are now included in Analysis of deadwood-associated organisms section

Statistical data analysis: I am puzzled by using square-root transformed data and Euclidean distance for the composition of enzyme families of CAZymes. Especially in combination with a PERMANOVA on bray Curtis distance, this seems a bit inconsistent to me. Can you explain, please?

Square root transformation specified as the Hellinger transformation. PERMANOVA testing omitted since its results were not included in the final version of the manuscript.

In this regard, would you mind providing values on the MDS-axes, please (Fig. 3B)?

Done

Figures and tables

Figure 2, caption: I suggest removing "other microbes" and give "Ascomycota and bacteria" directly; no need for information in brackets.

Brackets removed from caption of Fig 2

Figure S1: Maybe this is me, but it took me a moment to understand that the bar and square colors are the same both rows... Could you move up both legends and indicate "bars" and "squares" as legend title, please?

Figure S1 was modified as recommended

December 13, 2020

Dr. Petr Baldrian
Institute of Microbiology of the Czech Academy of Sciences
Laboratory of Environmental Microbiology
VÍdeňská 1083
Praha 4 14220
Czech Republic

Re: mSystems01078-20R1 (Complementary roles of wood-inhabiting fungi and bacteria facilitate deadwood decomposition)

Dear Dr. Petr Baldrian:

Your manuscript has now been accepted, and I am forwarding it to the ASM Journals Department for publication. I trust you to implement the minor issues identified by reviewer 2 in the proofs and to share sample metadata in an appropriate format.

For your reference, ASM Journals' address is given below. Before it can be scheduled for publication, your manuscript will be checked by the mSystems senior production editor, Ellie Ghatineh, to make sure that all elements meet the technical requirements for publication. She will contact you if anything needs to be revised before copyediting and production can begin. Otherwise, you will be notified when your proofs are ready to be viewed.

Sincerely,

Karoline Faust
Editor, mSystems

Journals Department
Phone: 1-202-942-9338